# Sensitivity Validation of EWOD Devices for Diagnosis of Early Mortality Syndrome (EMS) in Shrimp Using Colorimetric LAMP–XO Technique

**DOI:** 10.3390/s21093126

**Published:** 2021-04-30

**Authors:** Kreeta Sukthang, Jantana Kampeera, Chakrit Sriprachuabwong, Wansika Kiatpathomchai, Eakkachai Pengwang, Adisorn Tuantranont, Wishsanuruk Wechsatol

**Affiliations:** 1Department of Mechanical Engineering, King Mongkut’s University of Technology Thonburi, 126 Prachautid Rd., Bandmod, Thungkru, Bangkok 10140, Thailand; kreeta.suk@gmail.com; 2Bioengineering and Sensing Technology Research Team, National Center for Genetic Engineering and Biotechnology (BIOTEC), National Science and Technology Development Agency (NSTDA), Khlong Nueng, Khlong Luang, Pathum Thani 12120, Thailand; jantana.kam@biotec.or.th (J.K.); wansika@biotec.or.th (W.K.); 3Graphene and Printed Electronics for Dual-Use Applications Research Division (GPERD), National Science and Technology Development Agency (NSTDA), Khlong Nueng, Khlong Luang, Pathum Thani 12120, Thailand; chakrit.sriprachuabwong@nectec.or.th (C.S.); adisorn.tuantranont@nectec.or.th (A.T.); 4Institute of Field Robotics, King Mongkut’s University of Technology Thonburi, 126 Prachautid Rd., Bandmod, Thungkru, Bangkok 10140, Thailand; eakkachai@fibo.kmutt.ac.th

**Keywords:** droplet manipulation, detection sensitivity, colorimetric LAMP–XO, electrowetting-on-dielectric (EWOD), lab-on-a-chip (LOC)

## Abstract

Electrowetting-on-dielectric (EWOD) is a microfluidic technology used for manipulating liquid droplets at microliter to nanoliter scale. EWOD has the ability to facilitate the accurate manipulation of liquid droplets, i.e., transporting, dispensing, splitting, and mixing. In this work, EWOD fabrication with suitable and affordable materials is proposed for creating EWOD lab-on-a-chip platforms. The EWOD platforms are applied for the diagnosis of early mortality syndrome (EMS) in shrimp by utilizing the colorimetric loop-mediated isothermal amplification method with pH-sensitive xylenol orange (LAMP–XO) diagnosis technique. The qualitative sensitivity is observed by comparing the limit of detection (LOD) while performing the LAMP–XO diagnosis test on the proposed lab-on-a-chip EWOD platform, alongside standard LAMP laboratory tests. The comparison results confirm the reliability of EMS diagnosis on the EWOD platform with qualitative sensitivity for detecting the EMS DNA plasmid concentration at 10^2^ copies in a similar manner to the common LAMP diagnosis tests.

## 1. Introduction

Early mortality syndrome (EMS), also named acute hepatopancreatic necrosis disease (AHPND), is an acute severe liver disease in Pacific white shrimps, *Litopenaeus vannamei*. The mortality of the disease can be as high as 90% within 35 days in severely infected ponds [1]. *Vibrio parahaemolyticus* (VP) bacteria, spreading from the gastrointestinal tract to hepatopancreas tissues, was identified as the cause of the disease. EMS was first discovered in 2010, spreading across shrimp ponds in Southern China, and later affecting the shrimp business throughout the Southeast Asia region, causing an estimated loss of more than one billion USD yearly since 2012 [2].

The polymerase chain reaction (PCR) technique has been widely adopted for the early detection of pathogens in shrimp ponds [3]. The PCR technique is well-known for its capacity for exponential DNA amplification by thermal cycling of DNA samples using two main reagents, primers and DNA polymerase. Coupled with gel electrophoresis techniques, either by agarose gel electrophoresis or polyacrylamide gel electrophoresis, the so-called “PCR method” requires about 30 to 40 rounds of thermal cycling, in which all processes must be carried out in a standard laboratory, and around one day to deliver the diagnosis results to the shrimp farms. Meanwhile, quantitative polymerase chain reaction (qPCR) is used for the quantitative measurement of a normalized reporter value (Rn value) to identify the specific signal generated from a given set of PCR conditions under close monitoring. The qPCR TaqMan probe was adopted for shrimp pond pathogen detection in laboratory bioassays, with a reported detection sensitivity of around 10^2^ copies [4].

The loop-mediated isothermal amplification method (LAMP) has been adopted as an alternative method for DNA amplification and detection, with less processing time and simpler procedures than the conventional PCR and qPCR methods. The LAMP method relies on Bst DNA/RNA polymerase enzymes, which react at isothermal temperatures [5], thus eliminating the thermal cycling process required for typical PCR and qPCR methods. The LAMP method also allows visible detection, which is convenient for field test applications. For example, magnesium pyrophosphate can be used to induce the sedimentation of reaction products [6], fluorescent substances can be introduced for ultraviolet (UV) light detection [7], and simple methods such as the introduction of pH-sensitive dye can be applied to change the color of products for visible detection [8]. Table 1 shows various examples of LAMP applications and the required quantities of reactants.

To encourage more frequent use of the LAMP method and to ensure that reliable detection results and insignificant contamination are achieved in field test applications, lab-on-a-chip platforms (LOC), relying on various microfluidic techniques, have been applied and can be found in a variety of studies [14,15,16,17,18,19,20,21,22,23]. Table 2 shows examples of successful applications of LOC platforms in chemical and biochemical detection. In this work, the so-called microfluid technique electrowetting-on-dielectric (EWOD)—sometimes known by its more common name, “digital microfluidics”—was selected for creating LAMP–LOC platforms for the field detection of EMS in shrimp ponds. Today, EWOD devices are commonly used for microdroplet manipulation, i.e., transporting, splitting, mixing, and dispensing [24,25,26], and are quite suitable for manipulating the chemical reactants involved in LAMP detection, as mentioned in Table 1 and Table 2. 

The required quantities of chemical reactants for LAMP detection are normally between 1 and 25 μL. To prevent contamination from the surroundings, the closed-typed EWOD platform is composed of eight layers, i.e., substrate, bottom electrodes, bottom dielectric, bottom hydrophobic layer, medium substance, top hydrophobic layer, top electrodes, and a transparent lid, as shown in Figure 1, which shows the configuration of EWOD device selected for creating LAMP–LOC platforms. The working principle of EWOD devices is that, by applying an induced electrical field between two-adjacent electrodes on the bottom substrate to disturb the equilibrium of droplet surface tension, the electromotive force then causes the droplets to change their shape and move accordingly [27], as shown in Figure 2. In EWOD devices, droplet manipulation is performed on a flat plate and does not require a moving part, such as a micro-pump or any other complex mechanical configuration involving micro valves or micro piping networks; therefore, contamination resulting from assembly and operation can be avoided. Due to their ability to precisely manipulate liquid droplets combined with their compact configuration, EWOD devices are commonly used for proportioning chemical compounds in various reaction processes [28].

In this work, a LAMP–LOC platform was created and proposed for utilization in the field detection of EMS in Pacific white shrimp ponds. The configuration of the LOC is clearly presented. The colorimetric LAMP assay with pH-sensitive xylenol orange (LAMP–XO), which allows visible observation detection throughout the process, was applied to create the proposed LAMP–LOC platform. The sensitivity of detection is validated by observing the limit of detection (LOD) of the LAMP–XO diagnosis technique performed on the proposed LAMP–LOC platform in comparison with the LOD of the standard LAMP laboratory tests.

## 2. Sample Preparation

To validate the detection sensitivity and accuracy of the proposed LAMP–LOC platform in comparison with the common LAMP laboratory tests for EMS detection, the chemical compounds must be prepared beforehand in a contamination-free laboratory, as shown in Table 3. The purified EMS DNA samples were prepared following the recommendations made by Arunrut et al. [29]. Four reference EMS DNA plasmids with four different concentrations at 10^2^ copies, 10^3^ copies, 10^4^ copies and 10^8^ copies were prepared by diluting the purified EMS DNA samples in sterile distilled water (SDW). LAMP reaction mixtures consisting of each reference EMS DNA plasmid, 0.2 µM each of forward outer primer (F3) and backward outer primer (B3), 2 µM each of forward loop primer (LF) and backward loop primer (LB), 2 µM each of forward inner primer (FIP) and backward inner primer (BIP), 1× buffer for LAMP dye (pH 8.5), 1.2 mM dNTPs (Thermo Fisher Scientific, Waltham, MA, USA), 6 mM MgSO_4_ (New England Biolabs, Ipswich, England), 0.4 M betaine (Sigma Aldrich, St. Louis, MI, USA), 8 U Bst 2.0 WarmStart DNA Polymerase and 0.12 mM of pH-sensitive dye (xylenol orange; XO) (Sigma Aldrich, St. Louis, MI, USA) were prepared carefully. The volume of EMS DNA plasmids for standard laboratory tests was set at 2 μL, but due to the limit of the droplet size that could be handled by the LAMP–LOC platform with the proposed dimensions outlined in Section 3.1, the droplet sizes of EMS DNA plasmids were limited to 3 μL for LAMP–LOC platform testing. All primers were bought from Bio Basic Inc., Canada, and the details of primer sequences are reported in Table 4.

The LAMP reaction must be performed at the standard isothermal testing temperature of 64 °C for 75 min. Reaction mixtures without any DNA and with an EMS DNA plasmid concentration of 10^8^ copies must be prepared prior to testing to serve as the negative control and the positive control for this validation test.

## 3. Electrowetting-on-Dielectric (EWOD) Platform Fabrication

### 3.1. Simple Fabrication Method of EWOD Lab-on-a-Chip (LOC) Components

There is a strong relationship between the droplet sizes and the EWOD geometry. R. Malk et al. [30] produced EWOD electrodes based on fabrication techniques in the integrated circuit (IC) industry, in which EWOD devices are capable of manipulating droplets as small as 0.1 µL. Jie Gao et al. [31] utilized the photolithography fabrication technique in a cleanroom to produce the bottom electrode layers, which can control droplets as small as 0.7 µL. Chunqiao Li et al. [32] showed that plate-through-hole print circuit board (PTH-PCB) fabrication, which is a technique for fabricating multiple-layer printed circuit boards, could produce EWOD electrodes capable of controlling droplets with sizes as small as 1 µL. Vandana Jain et al. [33] created electrode layers on a single-layer printed circuit board that could control droplets as small as 4.5 µL. Yufia et al. [34] utilized electronic screen-printing techniques for electrode fabrication and successfully produced EWOD devices capable of controlling droplets as small as 10 µL. When comparing the capacity to manipulate droplets of EWOD devices fabricated based on different techniques, as described in Figure 3, and the droplet sizes required for performing DNA/RNA sequencing and detection, as shown in Table 1, PTH-PCB seems to be the most appropriate method for mass production due to its production cost and production reliability, which are important factors for field test applications. A comparison of the fabrication costs and fabrication limits for each electrode fabrication method is shown in Table 5, where PCB, PTH-PCB and screen-printing methods seem to provide cost competitiveness for LAMP–LOC platforms in field test applications. Therefore, PTH-PCB fabrication was utilized to produce the bottom electrode layer of LAMP–LOC platforms in this work.

With the aim of providing accessibility to LAMP–LOC platforms for the public, an unconventional fabrication procedure is proposed based on common materials from the electronic and thin-film markets. In Figure 4, a non-complexed fabrication procedure is proposed, composed of PTH-PCB bottom layer fabrication, film laying for producing the bottom dielectric/hydrophobic layer and in-house fabrication of the top lid. Compared to the conventional in-house cleanroom photolithography procedure and the screen-printing procedure, the proposed film-laying method eliminates the time consumption for dielectric and hydrophobic spin-coating processes and reduces the cost of the fabrication of dielectric and hydrophobic layers on top of the bottom plate, making the fabrication simpler than other conventional fabrication procedures. The PTH-PCB method, which allows users to produce multi-layer print circuit boards, can eliminate the complexity of circuit lines and allow users to create more complex electrode layouts than any other electrode fabrication techniques. The lid or the top plate, which is made of a glass slide coated with transparent conductive materials and covered with hydrophobic film such as Teflon-AF or the proposed PFC161V chemical matter, can be preordered or fabricated in-house.

#### 3.1.1. Electrode Design

The droplet manipulation inside a closed-typed EWOD device relies mainly on the impact of the disturbing phenomena of the electrical field on the equilibrium of droplet surface energy. By applying electrical potential to the adjacent electrode next to the droplet, the surface energy equilibrium of the droplet is disturbed, thus causing the contact angle to change, and drawing the droplet forward in order to maintain the state of equilibrium, as shown in Figure 2. To precisely control the droplet, the electrode geometry and configuration of EWOD devices must be designed specifically to match the droplet size.

##### Electrode Sizing

The size of the bottom electrode w and the spacer distance L between the top plate and the bottom plate in Figure 1 are critical parameters and are strongly related to the minimum size of the liquid droplets that the designed EWOD device can manipulate. The equation relating the controlled droplet volume to the electrode size w and the distance between the plates L can be described by the modified volume of a cut sphere, as described in Equation (1)
(1)$$V_{f}=\frac{L}{12}\pi \left(3w^{2}+2L^{2}\right)$$
where V_f_ is the volume of liquid droplets in a closed-type EWOD [mm^3^], *w* is the size of the bottom electrodes [mm] and *L* is the distance between the top plate and bottom plate [mm]. Figure 5 shows the volume of the controlled liquid droplets according to the size of the electrode w at the specified distance *L*. In this work, the spacer distance *L* was fixed at 1 mm, because non-conductive material with a thickness of 1.0 mm, such as a glass slide, was available. The electrode sizes (*w* × *w*) are 2.0 × 2.0 mm and 5.5 × 5.5 mm. The gap between two adjacent electrodes must be as small as possible, depending on the fabrication technique, i.e., for the plated-through-hole print circuit board (PTH-PCB) fabrication technique, the gap size is limited to 0.15 mm. These electrode dimensions allows the EWOD device to control the minimum volume of droplets at the approximated sizes of 3 µL and 24 µL, respectively.

##### Electrode Pattern Design

Closed-type EWOD devices provide the ability to transport, mix, split, merge, and dispense droplets, as shown in Figure 6. The number of electrodes required for mixing, splitting, and merging are three adjacent electrodes, while the number involved in the droplet dispensing process is four adjacent electrodes, including a reservoir electrode. During droplet transportation, electrical potential is applied to the adjacent electrode in the transporting direction. Similarly, electrical potential is applied to array electrodes in the back-and-forth direction at high switching frequency to shake and to homogenously mix droplets. When electrical potential is applied to the frontal and backward electrodes simultaneously and equally, the droplet can be split in half. Merging can be achieved by applying electrical potential to the electrode situated between two droplets. Dispensing is more complicated than the other droplet manipulation processes. Firstly, the droplet must be drawn out of the reservoir by applying electrical potential to three adjacent electrodes, as shown in Figure 6; later, the electrical potential applied to the two middle electrodes must be removed while maintaining the applied electrical field on both the reservoir and the end electrode in order to cut the droplet. The size of the dispensed droplet depends on the electrode size and the gap between the top and the bottom plates in Equation (1).

By applying the droplet manipulation patterns in Figure 6, we can create an EWOD electrode pattern capable of performing the colorimetric LAMP–XO detection technique on a LOC platform, as shown in Figure 7, where the reservoir that contains the reference EMS DNA plasmid can dispense droplets of the main chemical reactant in four directions simultaneously, allowing the transported droplets of the reference EMS DNA plasmid to merge and to mix with the applied LAMP–XO premix at each testing station to complete the preparation of LAMP reaction mixtures for the isothermal DNA amplification process. The proposed design of the EWOD platform aims to allow the colorimetric LAMP–XO detection technique to be performed conveniently during field tests with reliable test results, regardless of contamination concerns.

#### 3.1.2. Plate-through-Hole Print Circuit Board (PTH-PCB) Fabrication of Electrode Layout on the Bottom Substrate

For simplicity, low fabrication costs and reliable production, the PTH-PCB technique was selected to create the bottom electrode layer and the bottom plate substrate, which is much more convenient compared to the convention photolithography method and screen-printing method, where a glass slide is required as a substrate in order to imprint the electrode layout. The PTH-PCB technique also provides the opportunity to create multi-layer PCB up to 30 layers, which allows LOC designers to eliminate wiring and to create more complex testing procedures on any LOC platform. In Figure 8, double-layer PTH-PCB was designed and fabricated to serve the colorimetric LAMP–XO functional test, as proposed in Figure 7. The left figure shows the electrode layout on top of the bottom plate and the right figure shows the shadow of the wiring layer under the coated soldering mask.

#### 3.1.3. Fabrication of the Dielectric and Bottom Hydrophobic Layers

Polytetrafluoroethylene (PTFE) film in Figure 9, which is an electrical insulator with hydrophobic characteristics, was selected and proposed to replace both the conventional dielectric coating of Parylene-C, SU-8 or SYLGARD™ 184 Silicone Elastomer Kit (PDMS) and the hydrophobic coating of common Teflon-AF to complete the fabrication of the bottom EWOD layer. PTFE film with a thickness of 40 microns can be applied directly on top of the bottom PTH-PCB substrate. Mineral oil is recommended to provide adhesion between the PTH-PCB substrate and the PTFE film, as well as to prevent air bubbles forming between both layers.

#### 3.1.4. Spacer Layer Set-Up

The spacer layer of the EWOD device was created by using a thin-film layer with 1 mm thickness, placed around the rim of the bottom EWOD plate to create a 1 mm distance between the top plate and the bottom plate, according to the proposed design in Section 2. Later, silicone oil was used as a medium to fil the void space and to prevent the sample droplets from breaking down during droplet manipulation or evaporating during endothermic testing under heating conditions. Silicone oil also helps to maintain the sample temperature under isothermal conditions during the DNA amplification process.

#### 3.1.5. Top Plate Fabrication

The top plate (or lid) of closed-type EWOD devices is composed of fluorine-doped tin oxide (FTO) glass coated with a hydrophobic substance. FTO glasses are glass slides precoated with transparent conductive film, which are suitable for use as conductive grounds for EWOD devices. The transparency of FTO glasses allows us to observe the behavior of droplets inside EWOD devices. Before placing the FTO glass on the spacer, as shown in Figure 10, the FTO glass must be coated with a hydrophobic substance. The commercial FluoroPel@ PFC 1601V product was selected and proposed to replace Teflon-AF due to its better durability under applied electrical potential. Small holes with a diameter of 1 mm can be drilled through the top plate at any specified sample loading positions. Micropipettes can be used to control the droplet size of the loaded samples through the drilled holes.

FluoroPel@ PFC 1601V can introduce a droplet contact angle of up to 115°, which is crucial for the transportation of droplets inside EWOD devices. The larger the contact angle, the smaller voltage the required voltage is. The thin film of FluoroPel@ PFC 1601V, with its thickness of roughly 200 nanometers on top of the FTO glass, can be fabricated by utilizing the spin-coating process at 1000 rpm for 30 s and then baking in a hot air oven at 180 °C for 15 min.

### 3.2. Control Elements of LAMP–LOC Platform

The LAMP–LOC platform must be able to create isothermal DNA/RNA amplification conditions in the range of 64 ± 1 °C and to manipulate droplets accurately in terms of dispensing, splitting, transporting, merging, and mixing. In this section, the temperature control elements and the droplet manipulation control elements are described.

#### 3.2.1. Isothermal Heating and Temperature Control Elements

In order to enable reliable and trustworthy DNA/RNA amplification based on the LAMP technique, the isothermal heating condition must be confirmed. A 150 kW heating box and the Primus TMP-48 P-N-A PID temperature controller were installed to heat up the entire EWOD bottom plate, while transmitting heat to the sample droplets isothermally, as shown in Figure 11. An aluminum plate was installed as the physical heat capacitor to help prevent temperature fluctuations during tests. The experimental results confirmed that, by maintaining the core temperature of the aluminum plate at 84 ± 1 °C, the temperature of the samples at four test stations can be maintained accordingly at the target temperature of 64 ± 1 °C, as reported in Figure 12. The target temperature was reached within 1000 s after applying heat from the electrical heater to the aluminum plate.

#### 3.2.2. Droplet Controller

In this work, AC electricity was used to disturb the surface energy of the droplets inside closed-typed EWOD devices in order to alter droplets and change their shape, thus inducing droplet movement and manipulation. The droplet controller, with its major control equipment, was composed of a function generator, a power amplifier, a transformer, and a switching controller, as shown in Figure 13, where the left figure shows a schematic diagram of the controller connection, and the right figure shows the assembly of the LAMP–LOC platform. The function generator had a duty to generate alternating current (AC) sinusoidal signals at an amplitude of 10 Vp-p in a frequency range between 50 Hz and 5 kHz. The power amplifier was used to enlarge and fine-tune the amplitude of AC sinusoidal signals, while maintaining the supplied electrical frequency from the function generator. The transformer was used to enlarge the amplitude of the AC sinusoidal signals between 300 and 700 Vrms, which was a sufficient range for manipulating droplets inside the closed-type EWOD devices fabricated following the proposed procedure. The switching controller (see circuit diagram in Figure 14) worked as the signal distributor to supply the electrical potential via relays to each electrode in order to induce droplet movement. An Arduino MEGA 2560 R3 microprocessor was selected to govern the supply of electrical potential to any specific electrode locations via the relays. Mineral oil or silicone oil filled the void space and acted as a medium between the bottom plate and the top plate. During LAMP–XO testing on the LAMP–LOC platform, the suitable AC signals were set to 700 Vrms and 1 kHz. An AC signal frequency higher than 1 kHz would not affect the performance of droplet manipulation on any EWOD platforms [20].

## 4. Qualitative Sensitivity Validation of LAMP–XO EWOD Platforms

Typically, the LAMP method was performed in a thermal cycler (Applied Biosystems™: SimpliAmp™ thermal cycler), in which thermal cycling for DNA/RNA amplification could be adjusted to provide isothermal DNA/RNA amplification according to the specified testing procedure. For EMS detection, 110-μL LAMP–XO premix and 5-μL Bst polymerase must be homogenously mixed in a vortex mixer and a centrifuge for 10 s. The mixed compound was then placed in 5 microtubes, each containing 23 μL of the substance. During the sensitivity validation test, carried out in order to observe the limit of detection (LOD), the 2-μL EMS DNA plasmid samples with concentrations of 0 copies, 10^2^ copies, 10^3^ copies, 10^4^ copies and 10^8^ copies were added to each microtube. The 0-copy EMS DNA plasmid sample was used as the negative reference and could be prepared using SDW. The microtubes were then placed into a vortex mixer and a centrifuge to guarantee homogenous conditions prior to being placed in a thermal cycler. In the thermal cycler, DNA/RNA amplification was maintained at 64 °C for 75 min. After the isothermal amplification process, the color of the positive samples supposedly turned from violet to yellow, as shown in Figure 15. The sensitivity of EMS detection by the standard LAMP tests was limited to 10^2^ copies, in agreement with the work carried out in [29].

An experimental test for EMS detection on the LAMP–LOC platform was performed to observe the sensitivity and repeatability of the proposed LAMP–LOC platform. To prevent contamination on the LAMP–LOC parts that come into direct contact with the samples, a new and sterilized polytetrafluoroethylene (PTFE) film was applied to the bottom plate and the FTO top layer/lid was washed with AM9890 DNA Zap™ PCR DNA degradation solution. During the test, a homogenous mixture of 88-μL LAMP–XO premix and 4-μL Bst polymerase was equally divided and placed using a micropipette through the drilled holes at each mixing station, while 72 μL of DNA plasmid was placed at the inner dispensing station. The four different concentrations of EMS DNA plasmid samples selected, with concentrations of 0 copies, 10^2^ copies, 10^3^ copies and 10^8^ copies, were used to validate the sensitivity of the LAMP–LOC platform. For each plasmid concentration, 3 μL plasmid samples were automatically and simultaneously dispensed and delivered to the four mixing stations, as shown on the top of Figure 16, where the red lines envelop the boundary of the EMS DNA plasmid droplets dispensed and delivered to the mixing stations. After homogenous mixing conditions were achieved at each mixing station, the droplets at each mixing station underwent an isothermal heating process at 64 ± 1 °C for 75 min to provide sufficient DNA/RNA amplification for EMS detection. Xylenol orange (XO) causes the color of samples to change from violet to yellow in the event of *Vibrio parahaemolyticus* (VP) infection. The experimental results show that the proposed LAMP–LOC platform can provide positive detection results at all four stations for plasmid concentrations of 10^2^ copies, 10^3^ copies and 10^8^ copies, as shown on the bottom of Figure 16, where the color change happens as early as 60 min for the high plasmid concentration of 10^8^ copies, and the color change happens clearly after 75 min for the lower plasmid concentration of 10^2^ copies and 10^3^ copies under the isothermal amplification process. In Table 6, the test results obtained using the standard LAMP–XO technique performed on a thermal cycler, and the LAMP–XO technique on the LAMP–LOC platform, were compared with the test results based on the standard LAMP procedure with gold nanoparticles as probes (LAMP–AuNP) and the standard LAMP procedure with agarose gel electrophoresis, described in the literature by Arunrut et al. [29]. The sensitivity and repeatability test confirmed that the proposed LAMP–LOC platform can provide similar sensitivity and reliability for early detection of EMS at 10^2^ copies as the other standard LAMP laboratory tests mentioned.

## 5. Conclusions

The LAMP–LOC platform was designed, fabricated with suitable and affordable materials, and tested with the aim of confirming the sensitivity and reliability of EMS detection. PTH-PCB was selected as a suitable electrode layer for the proposed LOC platform. The appropriate electrical control signal appeared to be AC at 700 Vrms and a frequency of 1 kHz in order to ensure the perfect manipulation of droplets on the proposed LAMP–LOC platform. A functional test was also carried out, which showed that the LAMP–LOC platform had a similar sensitivity (measured in terms of LOD) to the other LAMP laboratory tests described in this study. The LAMP–LOC platform can detect *Vibrio parahaemolyticus* (VP) infection at an EMS DNA plasmid concentration as low as 10^2^ copies.

The proposed LAMP–LOC platform is also capable of the simultaneous detection of multiple diseases, which provides an advantage over the other conventional LAMP laboratory testing procedures and is crucial for field test applications. For example, in the case of shrimp pond infections, four different LAMP–XO premixes for the detection of EMS/AHPND disease, white spot disease (WSD), hepatopancreatic microsporidiosis disease (HPM) and hypodermal/hematopoietic necrosis virus infection (IHHNV), which are common severe diseases in shrimps [36], can be applied to the four mixing stations, thus allowing the LAMP–LOC platform to diagnose four different types of shrimp diseases simultaneously using just one drop of a DNA sample.

In terms of the cost benefits, conventional PCR, qPCR and LAMP laboratory tests require thermal cyclers to enable thermal cycling amplification processes, while the simple heating box of the proposed LAMP–LOC platform can carry out similar tasks and eliminates the need for a thermal cycler costing USD 5000, instead utilizing a LAMP–LOC platform testing set that costs only USD 1700, as shown in Figure 13. Overcoming further challenges with regard to replacing the function generator and power amplifier with a signal-generating circuit board would significantly reduce the cost of the LAMP–LOC testing platform to within USD 350.

## Figures and Tables

**Figure 1 sensors-21-03126-f001:**
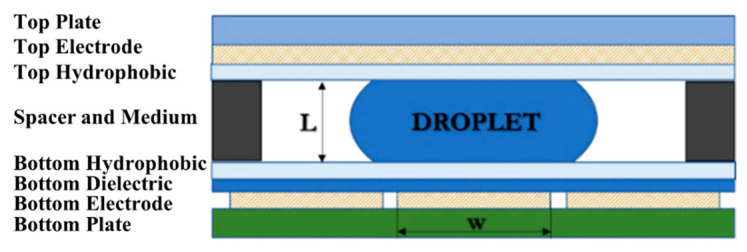
Configuration of a closed-typed EWOD device.

**Figure 2 sensors-21-03126-f002:**
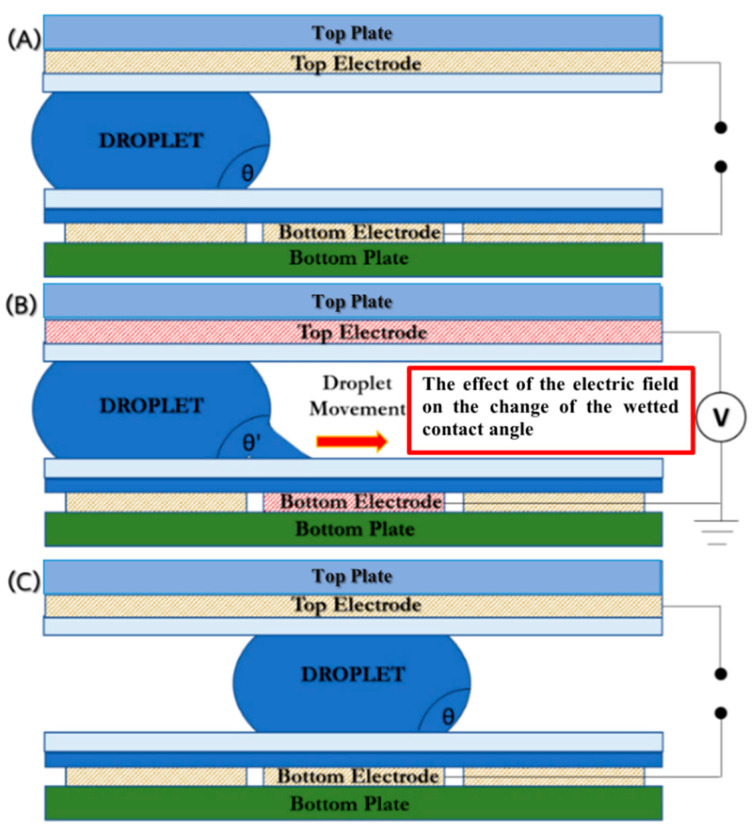
Effect of applied electrical potential on a droplet in a closed-type EWOD. (**A**) the droplet initial contact angle, (**B**) the change of contact angle due to the applied electrical field causing the droplet to move to the adjacent electrode, and (**C**) the droplet situated at the new position.

**Figure 3 sensors-21-03126-f003:**
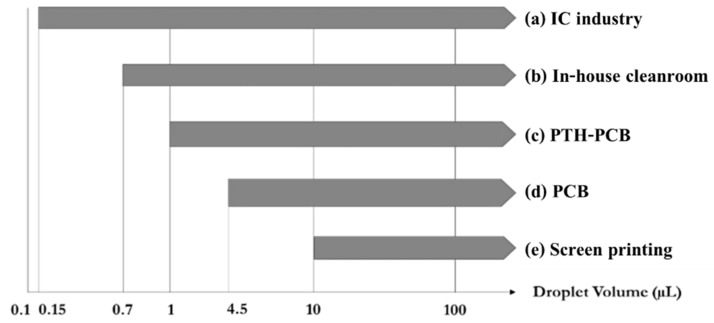
Possible droplet volume according to fabrication techniques. (**a**) R. Malk et al., (2011) [30]; (**b**) Jie Gao et al., (2015) [31]; (**c**) Chunqiao Li et al., (2018) [32]; (**d**) Vandana Jain et al., (2015) [33]; (**e**) M. Yafia et al., (2014) [3].

**Figure 4 sensors-21-03126-f004:**
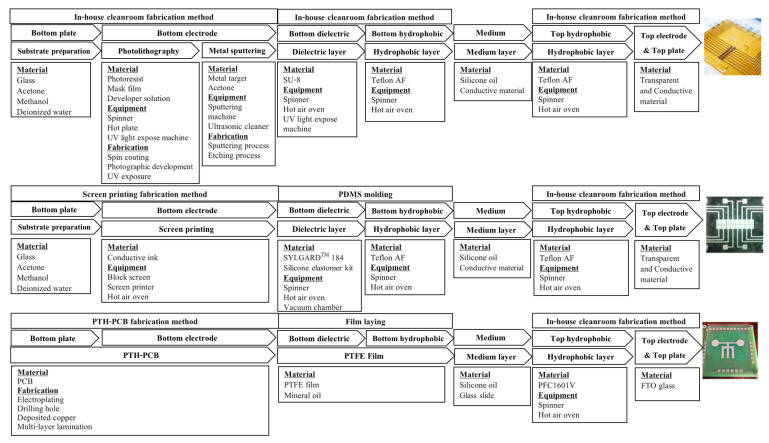
Comparison between the proposed bottom procedure and another conventional EWOD fabrication procedure.

**Figure 5 sensors-21-03126-f005:**
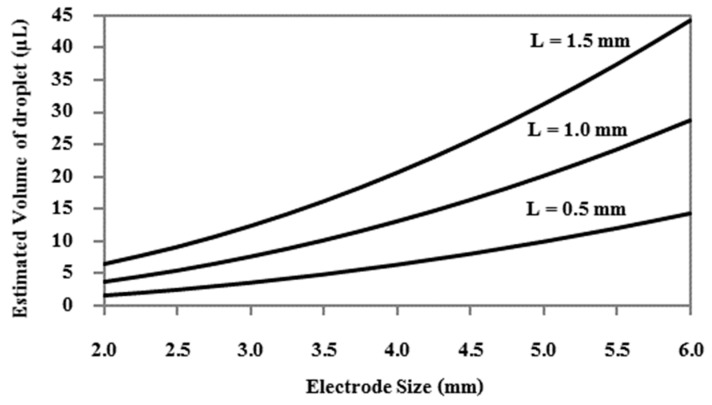
Relation of electrode size (*w*) and the estimated volume of liquid droplets in closed EWOD devices with the distance between the plates (*L*).

**Figure 6 sensors-21-03126-f006:**
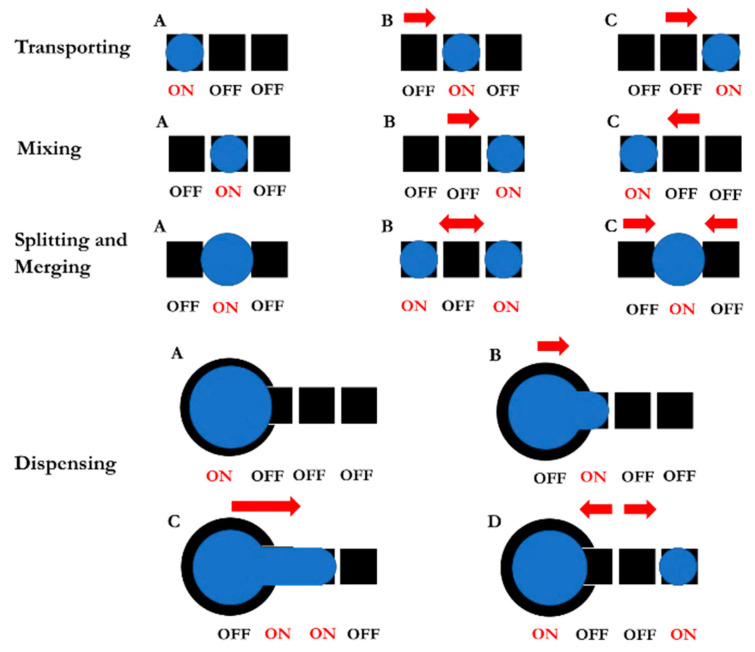
Characteristics of electrode design for controlling various liquid droplets, where the droplet controlling steps for each manipulation category are shown in alphabetical order.

**Figure 7 sensors-21-03126-f007:**
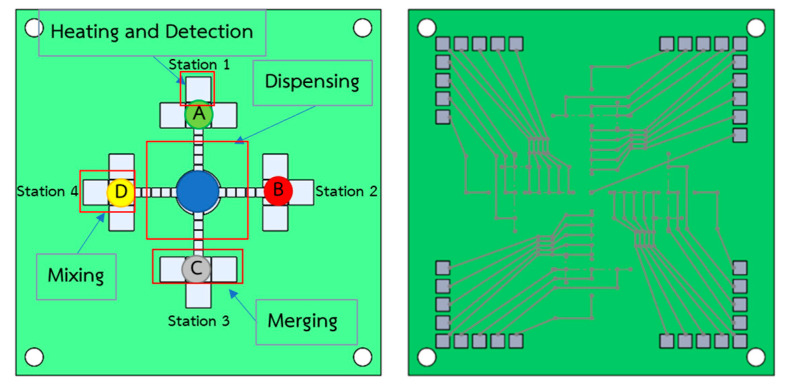
The bottom electrode layout design of the LAMP–XO EWOD platform.

**Figure 8 sensors-21-03126-f008:**
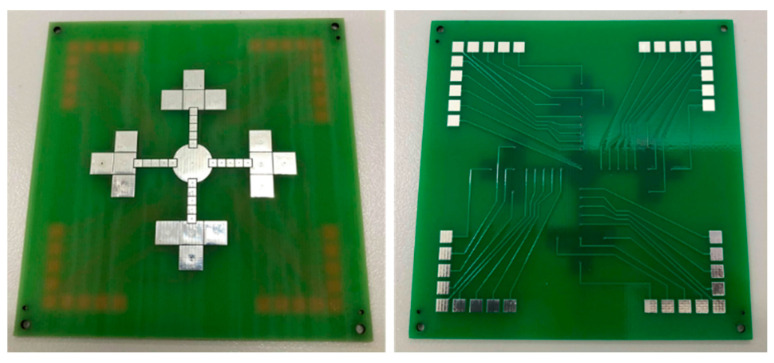
Double-layer electrode pattern fabricated by PTH-PCB method.

**Figure 9 sensors-21-03126-f009:**
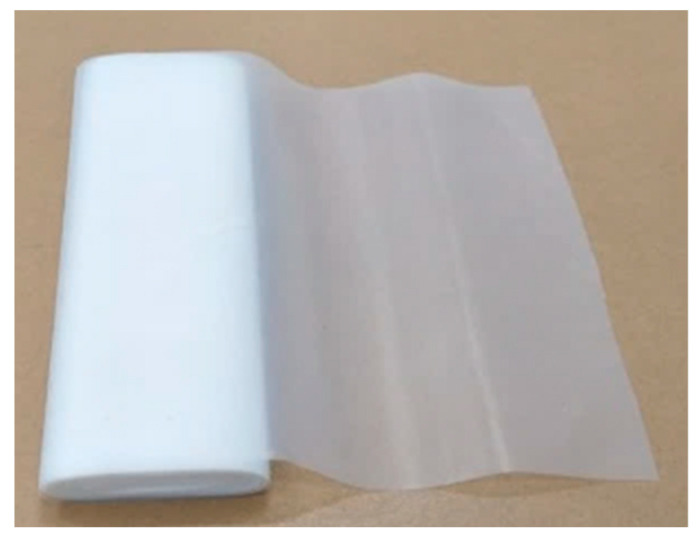
Polytetrafluoroethylene (PTFE) films with a thickness of 40 microns.

**Figure 10 sensors-21-03126-f010:**
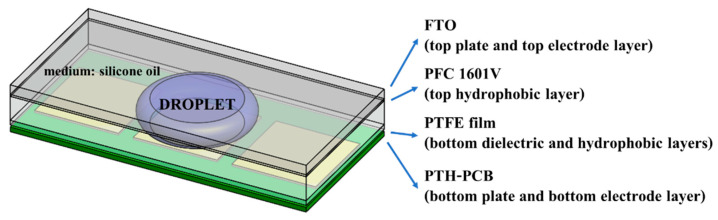
Components of closed EWOD device.

**Figure 11 sensors-21-03126-f011:**
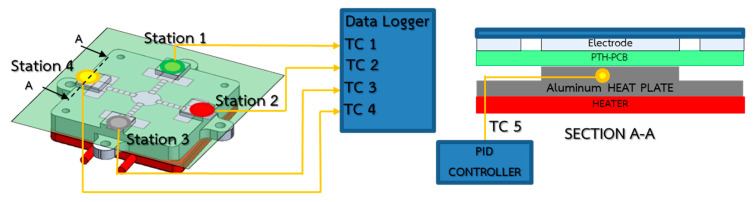
Installation of temperature control elements.

**Figure 12 sensors-21-03126-f012:**
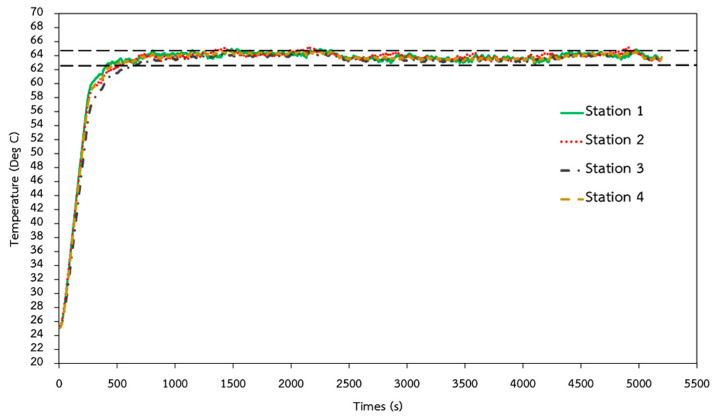
Temperature validation of sample droplets at four testing stations.

**Figure 13 sensors-21-03126-f013:**
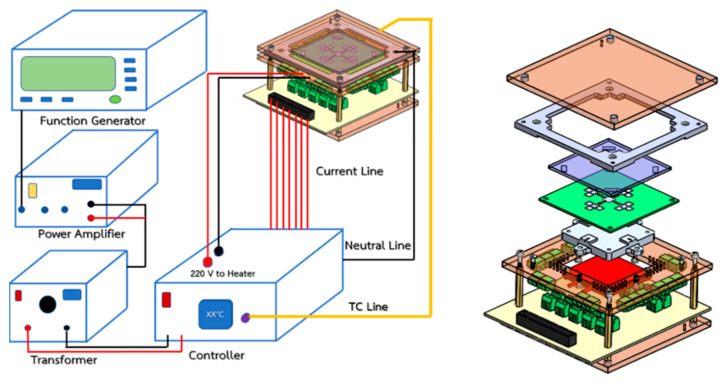
Schematic diagram of LAMP–LOC platform.

**Figure 14 sensors-21-03126-f014:**
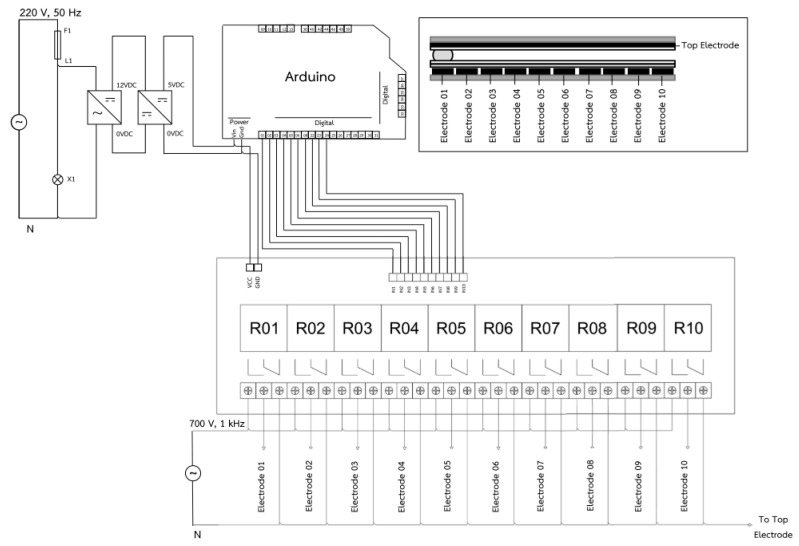
Driving circuit diagram of EWOD switching controller.

**Figure 15 sensors-21-03126-f015:**
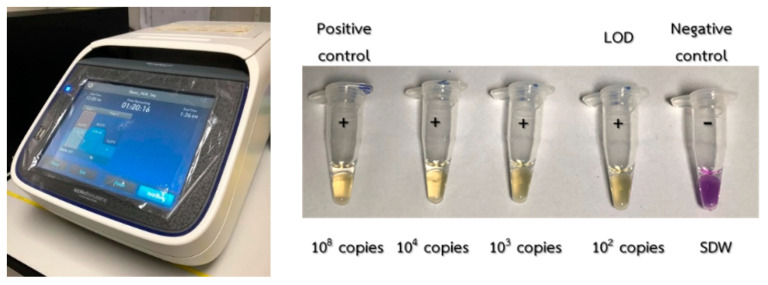
Sensitivity test results in thermal cyclers at a temperature of 64 °C for 75 min.

**Figure 16 sensors-21-03126-f016:**
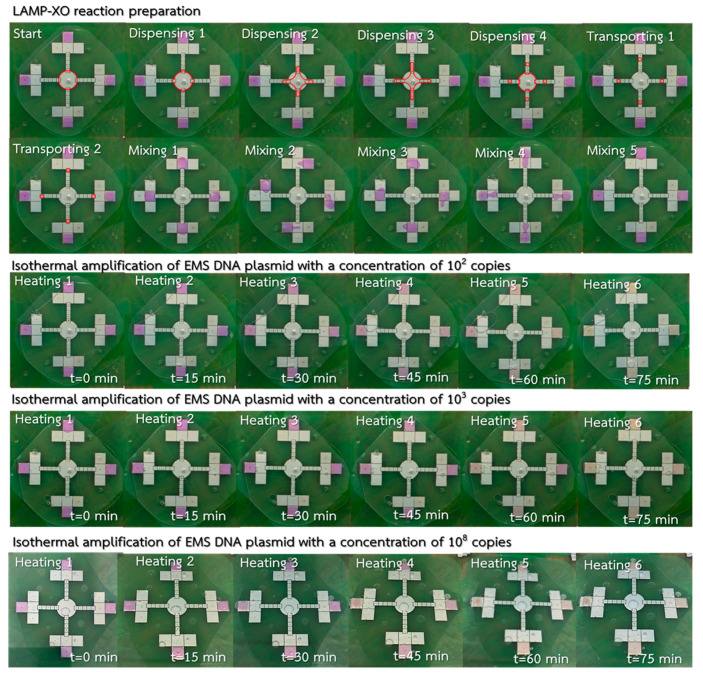
The experimental results, illustrating positive detection of EMS at 10^2^, 10^3^ and 10^8^ copies.

**Table 1 sensors-21-03126-t001:** Droplet sizes for loop-mediated isothermal amplification method (LAMP) DNA/RNA amplification and detection in a normal laboratory.

Amplification Technique	Detection Technique	Chemical Liquid for DNA Amplification	
		Substance	Volume [µL]
Reverse transcription loop-mediated isothermal amplification (RT-LAMP) [9]	Gel Electrophoresis	LAMP reaction mixture	25
		RNA Template	2
Loop-mediated isothermal amplification (LAMP) and polymerase chain reaction (PCR) [10]	Gel Electrophoresis and Visual Inspection	LAMP reaction mixture	25
		Reaction products	10
Loop-mediated isothermal amplification combined with DNA functionalized gold nanoparticles as probes (LAMP–AuNP) [11]	Colorimetric Detection	LAMP reaction mixture	25
		Product Volume	12
		DNA-functionalized AuNPs	5
		0.18 M MgSO4	2
Loop-mediated isothermal amplification (LAMP) [12]	Agarose Gel	Primer Mix	25
		Product	5
		Loading dye	1
Colorimetric loop-mediated isothermal amplification method with pH-sensitive xylenol orange (LAMP–XO) [13]	Colorimetric Detection	RNA Template	2
		LAMP reaction mixture	23

**Table 2 sensors-21-03126-t002:** Examples of lab-on-a-chip (LOC) applications and reactant droplet sizes.

Reference	Application	Procedure	Details of Droplet	
			Substance	Volume [µL]
Karuwan, C. et al. [14]	The electrowetting-on-dielectric (EWOD) electrochemical system for iodide droplet detection	Electrochemical detection	Buffer solution (i.e., hexahydroxy, HCL, etc.)	16
Grant, N. [15]	Electrowetting-on-dielectric (EWOD) for the extraction and isolation of DNA	Droplet preparation	DNA sample (blood)	0.26
			PNI buffer	1.3
			PE buffer	1.3
			Elution buffer	0.78
Karan V. et al. [16]	Real-time quantitative PCR microchip for influenza virus detection in human samples (nasopharyngeal swabs; throat swabs)	RT-PCR	PCR reagent mixture	5
			RNA	5
Dimov, N. et al. [17]	Droplet microfluidics (DMF) immunoassay platform for direct detection of pathogens	Real-time chemiluminescent measurements	Running buffer	2.5
			Pathogen antigens	2.5
			Pathogen samples	2.5
Taylor, J. B. et al. [18]	Hydrogel wax chip for malaria diagnosis	RT-PCR	Mastermixes	13
			DNA Template	8.5
Rival, A. et al. [19]	The isolation of a single cell, mRNA purification and gene expression analysis on EWOD platforms	mRNA extraction	qRT-PCR kit (i.e., buffer A, buffer B, elution buffer, etc.)	0.256–0.036
Fan, S. K. et al. [20]	Dielectrophoresis (DEP) and electrowetting-on-dielectric (EWOD) for concentration improvement of droplets containing neuroblastoma cells	Manipulated concentration of cell droplets	Neuroblastoma cells and polystyrene beads	0.45
Shah, P. et al. [21]	Single-cell nanotoxicity analysis using pDEP technique	Electrochemical detection	Trypan blue solution, PBS buffer, etc.	10
Jebrail, M. J. et al. [22]	Microfluidic system for quantification of amino acids in dried blood spots	DBS sample analysis	Blood samples	5
			Solvent	10
Barbulovic-Nad, I. et al. [23]	A cytotoxicity assay using Jurkat T-cells on a digital microfluidic platform	Cell-based assays	Jurkat T-cells (i.e., Tween 20, dyes, etc.)	0.15

**Table 3 sensors-21-03126-t003:** LAMP reaction component for early mortality syndrome (EMS) detection by standard laboratory methods and EWOD LOC.

Component	Final Concentration	Standard Laboratory Volume [μL]	LAMP–LOC Test Volume [μL]
100 μM FIP (EMS)	2 μM	0.5	0.5
100 μM BIP (EMS)	2 μM	0.5	0.5
10 μM F3 (EMS)	0.2 μM	0.5	0.5
10 μM B3 (EMS)	0.2 μM	0.5	0.5
100 μM LF (EMS))	2 μM	0.5	0.5
100 μM LB (EMS)	2 μM	0.5	0.5
10× low buffer for LAMP dye (pH 8.5)	1×	2.5	2.5
100 mM MgSO_4_	6 mM	1.5	1.5
10 mM dNTPs mix	1.2 mM	3	3
5 M Betaine (Sigma)	0.4 M	2	2
5 mM pH-sensitive dye (xylenol orange; XO)	0.12 mM	0.6	0.6
Sterile distilled water		9.4	9.4
8 U/µL Bst 2.0 WarmStart^®^ DNA Polymerase	8 U	1.0	1.0
EMS DNA plasmid		2	3
Total volume		25	26

**Table 4 sensors-21-03126-t004:** Primer sequences used for EMS detection by LAMP method.

Primers	Sequence (5′-3′)	Length (bp)
FIP-EMS	CGTTTGGTTCGACAGTCCAATTTTTATGAGTAACAATATAAAACATGA	48
BIP-EMS	GAGGCGGTCACAGAACTAGACATTTTCCCGTATTCTCAATGTCTACAC	47
F3-EMS	GTGCAATTTAATAGGAGAACATC	23
B3-EMS	GATTGGTAAGCTCCCCAC	18
LF-EMS	CGTGAGAATAGTCAGTT	17
LB-EMS	ACATACACCTATCATCCCGGAAG	23

**Table 5 sensors-21-03126-t005:** Comparison of integrated circuit (IC), plate-through-hole print circuit board (PTH-PCB), PCB, screen-printing, plastic molding, and in-house cleanroom technologies.

Manufacturing Technology	Fabrication Method	Electrical Compatibility	Typical Feature Size	Typical Device Size	Array Electrode Design	Typical Fabrication Cost
IC industry *	Thin-film planar and photolithography	2–10 layers	0.13–1 µm	1–100 mm	No	USD 31/cm^2^
PTH-PCB *	Electroplating and multi-layer lamination	2–30 layers	75–250 µm	1–100 mm	Yes	USD 0.021/cm^2^
PCB	Photolithography	1 layer	250 µm–10 mm	2–100 mm	No	USD 0.018/cm^2^
Screen-printing	Screen-printing	1 layer	350 µm–10 mm	2–100 mm	No	USD 0.0036/cm^2^
Polydimethylsiloxane (PDMS) molding *	Molding and soft lithography	1 layer	10 µm–10 mm	1–10 mm	No	Not sold commercially
In-house cleanroom *	Thin-film planar and photolithography	1–3 layer	2–100 µm	1–100 mm	No	USD 2/cm^2^

* Data extracted from Gong, J., and Kim, C. J. [35].

**Table 6 sensors-21-03126-t006:** Sensitivity test results of early mortality syndrome detection by LAMP–XO EWOD platform compared to the other standard LAMP tests.

Sample Type	LAMP Amplification and AGE Detection [29]	LAMP–AuNP [29]	LAMP–XO on Thermal Cycler	LAMP–XO on LAMP–LOC Platform	
				Results	Repeatability
Negative control (sterile distilled water)	Negative	Negative	Negative	Negative	5/5 (100%)
EMS DNA plasmid; 10^2^ copies (LOD—limit of detection)	Positive	Positive	Positive	Positive	5/5 (100%)
EMS DNA plasmid; 10^3^ copies	Positive	Positive	Positive	Positive	5/5 (100%)
Positive control (EMS DNA plasmid; 10^8^ copies)	N/A	N/A	Positive	Positive	5/5 (100%)

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
