# Peer review of "Sensitivity Validation of EWOD Devices for Diagnosis of Early Mortality Syndrome (EMS) in Shrimp Using Colorimetric LAMP–XO Technique"

_sensors, 2021, doi:10.3390/s21093126_

Round 1

Reviewer 1 Report

Sensors LAMP XO Review            

The English is extremely poor and detracts from the science presented.  For example in the Abstract on line 19:  Electro-wetting on Dielectric Device (EWOD) is a microfluidic technology ‘purposefully’ for manipulating liquid droplets.  Purposefully does not make sense and should be changed with ‘used’.  An English language speaker must check the text.  Grammar is poor throughout.  Within the abstract the only result stated is the final plasmid concentration; sensitivity is discussed (Ln 28) but the value is not stated.  In the abstract be specific about the control used as opposed to the equipment used, the authors state ‘Thermal cyclers’, PCR should be mentioned (line 26); this is a good example of poor grammar, the ‘thermal should not have a capital letter’.  The limit of detection (LOD) should be described as such and not detection limit (lines 29-30).

Acronyms are defined on multiple occasions, for example EMS defined on lines 24, 27 and 35.

Introduction

Any taxa should be given in italics (lines 36 and 38). 

Why do you not discuss qPCR and only PCR?

Lines 37-38:  Poor English, revise sentence, it should read something like ‘in severely infected ponds.’

Line 44:  Check where references are placed. 

Line 52-53:  Remove the full stop after Bst

Table 1 and 2 should be presented before Figures 1 and 2 as this is the order they are presented in the text. 

Line 88:  The authors describe calorimetric LAMP-XO as opposed to colorimetric.  This shows the poor care taken in preparing the manuscript.  There are too many mistakes in the paragraphs above to mention.  The authors do not describe what XO stands for.  Lab on a Chip, once given should be defined as LOC. 

Sample preparation

105: Ref position?

Lines 108-109:  Sigma Aldrich, remove e.

Table 3:  There is no explanation as to why the template DNA use in the EWOD device was 3uL compared to 2uL for the standard volume.  Why was 3uL used in the EWOD device?

Did the authors think to include more detail on the LAMP assay, number of primers etc. in introduction, this would help those unfamiliar with LAMP understand the need for 6 primers?

Where were the primer acquired?

From the abstract, the authors state the process is compared against standard laboratory test results.  However, on further reading, this appears to be LAMP in a PCR tube.  Did you compare with qPCR?

EWOD platform Fabrication

In equation 1 you refer back to Fig. 2, however in Fig. 2 you do not give dimensions.  A further diagram should be included with dimensions.

Line 139: Expand PTH-PCB. 

Why have the authors chosen PCB fabrication methods as opposed to photolithography?  Simple fabrication and fabrication steps should come before for the design stage as it is not possible to design something if you do not know the manufacturing technique and resolution of electrodes.  The authors should also discuss the fabrication method briefly in the introduction. 

Section 3.1 would be better placed after section 3.2.

Section 3.2 is of a much higher standard of English and is much easier to follow. 

Section 3.3.1:  Give a greater explanation of the electronic setup, what equipment was used, what was used to control the temperature?  Was this also using an Arduino?

Line 318:  Very simple error RNA

Line 331:  What thermocycler was used?

Line 353:  The transparent ‘rid’?

Table 5 shows ‘sensitivity’, this is subjective according to colour change.  How would the authors address this in the future?

Conclusion

No moving parts – could this be used in the ‘field’ as a lab on a chip?  The authors discuss low power consumption but do not quantify the power consumption in their unit. 

The paper lacks the detail required to repeat all experiments.  Whilst some sections such as the fabrication has good detail, other sections lack this.  Oversights such as missing out the name of the thermal cycler are poor practice.  Can LAMP really be used as the only control?

Author Response

The manuscript has been revised following the reviewer's comments. Please see the response to the reviewer's comments in the attached file. 

Reviewer 2 Report

The authors use EWOD to automate the dispensing, moving, mixing, and heating processes of the colorimetric LAMP-XO experiment. The method is new and practical and will potentially benefit POC workers and researchers in the LOC area.

However, the quality of the presentation must be improved before considering it for publication.

The main concerns are the missing information on the circuit design of the EWOD system and the LAMP primers/sequences. These materials can be added either in the main text or in the supplementary material.

A video that shows the operation of the EWOD device will be very helpful for potential readers. The main juice of this study is to show the automation of the experiment, static images in Figure 14 have some results but the evidence is not sufficient to prove that promise.

Consider including the following materials in the revision:

In Section 3.3.2, it is not clear how the Arduino MCU controls the switch of the 700 Vrms AC signal. In EWOD, many people use either sine waves or square waves. At high frequencies, the signal type shouldn’t affect the performance.

Not clear on how to sterilize the device and make it nuclease-free?

Very limited information about the driving circuit of the device was shown in the article. Either add them to the main text or in supplementary materials.

Would an oil environment help with reducing the evaporation during the 75min reaction?

The top plate covers the entire device. Use a smaller top plate and leave half of the reservoir uncovered can let the user add liquid to the reservoir without removing the top plate.

A video that shows the dispensing, mixing and moving of droplets will be helpful. Please consider adding a video to the supplementary materials.

In addition to the missing information, English errors must be corrected in the revision.

For example:

  1. Line 38, ‘severed infected ponds’ should be ‘severely infected ponds’.
  2. Line 42, ‘since the year 2012’ should be ‘since the year of 2012’.
  3. Still line 42, ‘PCR technique’, use either ‘The PCR technique’ or just use ‘PCR is well-known….’.
  4. Line 45, delete ‘relying of’.

These are just the errors found in the first paragraph of the Introduction section. There are English grammar errors throughout the article. Proofreading is a must for the re-submission.

Author Response

(The authors gave the same response as above.)

Round 2

Reviewer 1 Report

The authors have adjusted the manuscript according to the reviewers last comments.

Line 80:  The authors still refer to transparent rid, should this be traparent lid?

The overall standard of English is still poor and detracts from the science presented.  This must be read and corrected by an English language speaker.

Author Response

The manuscript has been revised following the reviewer's comments. Please see the response to the reviewer's comments in the attached file

Reviewer 2 Report

All the concerns have been addressed.

Please replace Fig. 14 (newly added) with a higher resolution one. 

Author Response

(The authors gave the same response as above.)
